# Qualitative exploration of factors associated with COVID-19 vaccination among pregnant women in Kenya

Violet Naanyu[1,2]*, Ferdinand Okwaro[1], Ingrid Gichere[3], Mandeep Sura[4], Prachi Singh[5], Berhaun Fesshaye[5], Marleen Temmerman[1,3]

**1** Centre of Excellence in Women and Child Health, Aga Khan University, Nairobi, Kenya, **2** School of Arts and Social Sciences, Moi University, Eldoret, Kenya, **3** Department of Obstetrics and Gynaecology, Aga Khan University, Nairobi, Kenya, **4** Pumwani Maternity Hospital, Nairobi, Kenya, **5** Department of International Health, Johns Hopkins Bloomberg School of Public Health, Baltimore, Maryland, United States of America

* violet.naanyu@aku.edu, vnaanyu@ampath.or.ke

## Abstract

The COVID-19 pandemic significantly disrupted societies worldwide, and COVID-19 vaccination has been identified as instrumental in the fight against the pandemic. While studies have examined uptake of COVID-19 vaccination, more research is needed to gain a better understanding of the factors influencing pregnant women's decisions to get vaccinated. In this study, we used a cross-sectional descriptive qualitative study to explore the attitudes, beliefs, and behaviors related to COVID-19 vaccination during pregnancy in Kenya. A total of 50 in-depth interviews were conducted between October 2023 to March 2024 with a purposively drawn sample of 25 pregnant and 25 postpartum women during their clinic visits. Interviews were conducted by trained personnel, audio-recorded, transcribed into English, and coded using NVivo 12 software. Thematic findings were organized using the SAGE Vaccine Hesitancy Model categories to provide a comprehensive report of the factors perceived to influence COVID-19 vaccination among pregnant women in Kenya. There were similarities in perceptions reported by pregnant and postpartum women. Contextual influences included communication and media, policies, and geographic factors. Six individual and group influences were personal, family, and community experiences with vaccination, beliefs and attitudes about health and prevention, knowledge and awareness about COVID-19 vaccines, knowledge of the health system and trust in healthcare workers, perceived risks and benefits, and social norms. There was only one vaccine-specific influence (the scientific evidence on benefits versus risks), and five vaccination-specific issues, including mode of administration, design of the vaccination program, reliability and source of vaccine supplies, vaccination schedule, and the strength of healthcare workers' recommendations. The COVID-19 pandemic highlighted challenges of maternal vaccine acceptance, and this calls for more

**Data availability statement:** All relevant data are within the paper, and data will be made available to interested researchers on request. All requests can be made to the AKU Kenya Research Office, Aga Khan Institutional Scientific and Ethics Committee, Email: AKUKenya.ResearchOffice@aku.edu.

**Funding:** This work was supported by the Bill & Melinda Gates Foundation grant INV-016877 to R.Limaye/ R.Karron. The funders had no role in study design, data collection and analysis, decision to publish, or preparation of the manuscript.

**Competing interests:** The authors have declared that no competing interests exist.

research to increase understanding of factors associated with pregnant women's decision-making on maternal vaccines, such as the COVID-19 vaccination, which can help inform interventions aimed at supporting this population.

## Introduction

COVID-19 vaccines have greatly contributed to global health by combating morbidity and mortality. They are cost-effective public health measures that can prevent disease spread and reduce the disease burden on healthcare systems [1]. Pregnant women with COVID-19 face an increased risk for severe morbidity and mortality, adverse preterm delivery, pregnancy loss, and stillbirth, hence the need to study vaccine hesitancy in this population [2–5]. There is evidence showing that those who receive the full dose of the COVID-19 vaccine experience decreased risks, e.g., preterm births and adverse neonatal outcomes, hence the need to extend the vaccination services globally [6,7]. Furthermore, maternal booster vaccinations are associated with lower neonatal deaths and infant COVID-19 hospitalizations, signaling the importance of counseling pregnant women towards full and proper vaccination practices [7,8]. To counter the pandemic, multi-institutional efforts resulted in the mass production of multiple COVID-19 vaccines [9,10].

In Kenya, frontline health workers, teachers, police, and military were targeted in the first phase of the vaccination program that was rolled out in March 2021 [11]. In September 2021, priority groups for vaccination were expanded to include citizens aged 50 years and all adults with underlying medical conditions or disabilities [12]. Pregnant and breastfeeding women in Kenya remained ineligible to receive COVID-19 vaccines [13], even though pregnant women have increased complications from infection in pregnancy, and thus vaccination programs ought to prioritize them [3,4,14–16]. The World Health Organization (WHO) lists pregnant women as a sub-population of special interest and recommends receiving a single dose of the vaccine during each pregnancy [16]. In August 2021, the Kenyan Ministry of Health (MoH) provided guidance on COVID-19 vaccination for pregnant or breastfeeding women following a risk-benefit assessment and consultation with a healthcare provider. In December 2021, the Kenyan Director General of Health issued a letter to county health directors recommending COVID-19 vaccines in pregnancy, with no additional requirements or qualifications [13]. From June 2023, Kenya has recommended COVID-19 vaccination for all pregnant and breastfeeding women [17].

Despite the evidence of COVID-19 vaccine efficacy and safety, uptake remains a significant challenge in many parts of the world [18–22]. Vaccine decision-making remains a complex matter among pregnant and postpartum women [23]. The decision to vaccinate is associated with gravidity, COVID-19 knowledge, misinformation, vaccine safety risks during pregnancy and breastfeeding, and vaccine efficacy [13,24,25]. Furthermore, unclear policy guidance for vaccination of this population, inadequate health worker counselling, and assumptions of ineligibility during pregnancy and/or breastfeeding can all complicate decision-making for vaccination

among pregnant and postpartum women [13,26]. The WHO Strategic Advisory Group of Experts on Immunization (SAGE) defines vaccine hesitancy as the refusal or delay in vaccination acceptance despite vaccination services' availability [27]. The COVID-19 pandemic exposed the urgent need to address vaccine hesitancy and confidence, while appreciating that these factors are complex, context-specific, and vary across time, place, and vaccines [27,28].

Barriers to COVID-19 vaccination have been noted in Kenya and similar global settings. On the supply side, factors affecting vaccination include the types of vaccine products available, the capacity of the healthcare workforce, procurement, distribution, and cold-chain capacity, linguistic challenges, information and communication on vaccination, and policies on prioritization of at-risk groups [29,30]. Barriers such as inadequate vaccine supplies, crowded vaccination sites, and long distances to vaccination services discourage vaccination [31–33]. Common demand-side factors associated with hesitancy include limited COVID-19 knowledge and vaccine skepticism, fear of adverse events, concerns about vaccine safety and side effects, lack of trust in the pharmaceutical industry, and misinformation [24,33,34].

This paper is based on a mixed-methods study conducted in four countries (Brazil, Ghana, Kenya, and Pakistan) that aimed to understand how pregnant and postpartum women make decisions about COVID-19 vaccines [35]. It focuses on specific findings from the qualitative component that captured the attitudes, beliefs, and behaviours related to COVID-19 vaccination during pregnancy in Kenya. Study findings are organized into sub-themes within the SAGE Vaccine Hesitancy Model categories [33], which encompass contextual influences, individual and group influences, and vaccine and vaccination-specific issues.

## Materials and methods

### Ethics statement

This study was reviewed and approved by relevant ethics review boards; Johns Hopkins School of Public Health (Ref. IRB00020861), the WHO's Ethics and Research Committee (CERC.0193B), the Nairobi County Research and Development Committee (NCC/CS/RPD/84/2023), the Aga Khan Institutional Scientific and Ethics Committee (2023/ISERC-17), and the PMH ethics committee (PMH/CEO/76/0785/2023). A research permit was obtained from the Kenyan National Commission for Science, Technology and Innovation (NACOSTI/P/23/29152). Formal written consent was obtained from all study participants.

### Study design and setting

The data used in this paper were collected using a cross-sectional descriptive qualitative design that utilized in-depth interviews (IDIs). The study was conducted in two hospitals in Nairobi, the Aga Khan University Hospital Nairobi (AKUH-N) and Pumwani Maternity Hospital (PMH). Aga Khan University Hospital is a private teaching and referral healthcare facility that serves clients from the middle to high socioeconomic groups. The Department of Obstetrics and Gynaecology at AKUH-N attends to 200–250 patients per month with a broad case mix, managing pregnancy and childbirth complications, including those referred from other facilities within the East African Region. Pumwani Maternity Hospital, located in a lower-income area of Nairobi, is Kenya's biggest specialized health facility dedicated to maternal and newborn care, handling an average of 600 clients per month and serving clients from resource-poor neighbourhoods, including informal settlements.

### Study participants

Participants were women who were pregnant or recently pregnant (up to six weeks post-partum) visiting AKUH-N or PMH for antenatal care (ANC), post-natal care (PNC), and/or any other pregnancy-related visit. The study aimed to interview 50 women: 25 antenatal patients and 25 postnatal patients. Further, among those who were pregnant, equal samples of women from each of the three trimesters were the goal for recruitment. Both sites employed consecutive sampling of

women in the clinic waiting area, with the PMH site also including recruitment from the reception area to capture more first trimester patients, who were likely to turn away for lack of registration fees.

## Data collection

Data collection occurred between October 2023 to March 2024. Before the start of data collection, the study team participated in a three-day training session covering human participants' research ethics, among other topics. The study employed two female research assistants who had no prior relationship with the potential study participants. They were graduates in Public Health and pursuing Master's studies at the time of the project. Recruitment occurred at the ANC and PNC clinics of the two hospitals using the following criteria: 1) study interest, 2) age of 18 or older, 3) fluency in Kiswahili or English, and 4) knowledge of the COVID-19 vaccine. Eligible participants were provided information about the study, and if they still expressed interest, written informed consent was obtained.

In-depth interviews took on average 1 hour and were conducted in English or Swahili by two trained research assistants using semi-structured interview guides (S1 File), one for pregnant women and one for postpartum women. General questions captured the participant's locality, household composition, age, details of the current pregnancy, knowledge and prevention of COVID-19, community views on COVID-19, vaccination of family members, and government efforts to protect pregnant and postpartum women against COVID-19. The second part of the tool consisted of vaccination-related questions. The sub-topics included vaccines received and experience getting vaccinated during pregnancy/post-pregnancy; choice of vaccine; sources of information about the COVID vaccine for women who were pregnant/postpartum; the questions they asked about the vaccine and information received; decision-making influencers; availability of the vaccine; personal, family, and community views on whether pregnant/postpartum women should get the COVID vaccine; and challenges faced trying to get vaccinated.

All interviews were audio recorded, and field notes were used to document the venue, date, and details about participants. Data quality checks were run before analysis. To ensure the ethical safeguard of participants, audio recordings and transcripts were kept confidential, and all direct identifiers were removed before analysis and publication of any results.

## Data management and analysis

All fifty audio files were transcribed and translated by research assistants who were fluent in English and Swahili. Two coders immersed themselves in the data to gain a comprehensive understanding of the content and used NVivo 12 software to support data analysis. The data collection guides were used to develop an initial codebook, which was adjusted with emergent inductive codes. The findings were thematically analyzed and examined by category of participants (pregnant, postpartum; vaccination status). The themes were consequently organized as sub-themes within the SAGE Vaccine Hesitancy Model categories [36] to comprehensively report the factors that could potentially influence COVID-19 vaccination among pregnant women in Kenya.

The SAGE model (S1 File) was used to organize the study findings because it has three main categories of factors associated with vaccine hesitancy, which allowed the researchers to logically report the attitudes, beliefs, and behaviors related to COVID-19 vaccination during pregnancy in Kenya. The model's three factors include contextual influences, individual and group influences, and vaccine and vaccination-specific issues. Each of these categories has specific determinants, and some of them overlap. Only specific determinants from the model aligned with the themes noted in our study findings are discussed below. Furthermore, we report the factors associated with the third category in two separate sub-themes: vaccine-specific and vaccination-specific influences.

## Results

This study held 50 in-depth interviews with pregnant and postpartum women from Kenya (Table 1).

Table 1. Demographic and health characteristics of study participants.

| | Pregnant | Postpartum | Total |
|---|---|---|---|
| **Total participants (n)** | 25 | 25 | **50** |
| **Age** | | | **50** |
| Reported age range (years) | 20-41 | 19-41 | |
| Mean reported age (years) | 30 | 28 | |
| **Pregnancy trimester** | | | **25** |
| First (n) | 9 | – | |
| Second (n) | 9 | – | |
| Third (n) | 7 | – | |
| **Gravidity among postpartum women** | | | **25** |
| Primiparous (n) | – | 9 | |
| Multiparous (n) | – | 16 | |
| **COVID-19 vaccination status** | | | |
| Women who had received the COVID-19 vaccine | 17 | 20 | **37** |

The 50 participants ranged from ages 19–41 years and were split evenly between pregnancy and postpartum. Of the 25 pregnant participants, 9 were in the first trimester, 9 were in second trimester, and 7 were in third trimester. 40% were primigravid. Of the 25 postpartum participants, 36% were primiparous and 64% had more than one child. Thirty-seven of the 50 women interviewed had received the COVID-19 vaccine.

Study findings align with several factors from the SAGE model. They reveal three contextual influences, six individual and group-level influences, one vaccine-specific influence, and six vaccination-specific issues. These are discussed below and summarized in Table 2.

In the sources of the illustrative quotes provided, the A and P followed by a number represent participants from specific study sites: A for AKUH-N or P for PMH.

## Contextual factors

Three contextual influences from the SAGE model were relevant in this study (Table 2): 1) communication and media, 2) policies and politics, and 3) geographic factors.

Table 2. Representation of the SAGE model's determinants that can be associated with COVID-19 vaccination among pregnant women in Kenya.

| **1. Contextual factors** | • Communication and media<br>• Policies<br>• Geographic factors |
|---|---|
| **2. Individual and group factors** | • Personal and community experiences with vaccination<br>• Beliefs and attitudes about health and prevention<br>• Knowledge and awareness about COVID-19 vaccines<br>• Knowledge of the health system and trust in healthcare workers<br>• Perceived risks and benefits<br>• Social norms on the need for COVID-19 vaccines |
| **3. Vaccine-specific influences** | • Risk/benefit scientific evidence |
| **4. Vaccination-specific influences** | • Mode of administration<br>• Design of the vaccination program<br>• Reliability and source of vaccine supplies<br>• Vaccination schedule<br>• Healthcare workers' recommendations, knowledge base, and attitudes |

**Communication and media.** The sources of information reported by vaccinated pregnant and postpartum women included mainstream media (television and radio), government announcements on COVID-19, and social media (e.g., WhatsApp, Facebook, Instagram, or YouTube). The information received from the foregoing sources covered diverse topics, including COVID-19 symptoms, transmission, risks, prevention, and side effects of the vaccine.

"We were told about COVID-19 in the clinics, and they encouraged us to take the vaccine to protect ourselves and our families." (P23, Pregnant, Vaccinated)

"The government has done a good job… educating people on how to avoid COVID, such as wearing masks and sanitizing." (P08, Postpartum, Vaccinated)

The unvaccinated women reported the same three contextual influences from the model and similar sources of information. Additional findings from unvaccinated pregnant and postpartum women captured myths and rumors in their communities, implying the COVID-19 vaccine caused more harm than good, e.g., triggered infertility, death, and cancer. Other messages suggested COVID-19 and associated vaccination programs had hidden motives, e.g., population control and income generation.

"There were so many rumors [like] 'people will get infertile.'" (A09, Postpartum, Unvaccinated)

"Because they [community members] lack information, they are very scared of getting vaccinated. They think it's something that the government wants to generate income, or NGOs want to generate income." (A14, Pregnant, Unvaccinated)

"Not all of them [community members] are very literate. So, when COVID hit, rumors were that it's a way of the government reducing the population." (P26, Pregnant, Unvaccinated)

Some vaccinated participants reported messaging on the increased risk for pregnant women and information about taking extra precautions during pregnancy. Some messages included the benefits of the COVID-19 vaccine to both the mother and the baby, and that it was safe for pregnant women. Nonetheless, the information on increased risk for pregnant women and information about taking extra precautions during pregnancy was not universally known by our vaccinated and unvaccinated study participants

"I haven't heard of anything about what pregnant women should do with the COVID vaccine." (P25, Pregnant, Unvaccinated)

**Policies.** Government policies were reported by vaccinated women as associated with vaccination. These included policies and guidelines encouraging vaccination as a way to protect the most vulnerable populations from COVID-19 infection; government-driven COVID-19 prevention measures (e.g., vaccination, hygiene measures, and social distancing); community outreach programs (e.g., COVID-19 educational and mobile vaccination services); and COVID-19 vaccination requirements for work and travel.

"The government provided vehicles with PA systems and went around sensitizing people to go take the vaccine." (A11, Pregnant, Vaccinated)

"We had to get vaccinated because it was a requirement for international travel." (A22, Postpartum, Vaccinated)

**Geographic factors.** The last category of contextual influences is the geographic factors reported by pregnant and postpartum vaccinated women. Reports suggested that short travel to an information and vaccination site encouraged vaccination.

"They brought vaccines to the community and provided information about how to avoid COVID-19." (P08, Postpartum, Vaccinated)

Geographic factors reported by unvaccinated pregnant and postpartum women were related to costs associated with getting vaccinated, even though the service was provided at no cost. For instance, the fare to vaccination sites discouraged uptake of the service.

"The vaccine was free, but I had to budget for the transport. It wasn't much, but it was still a consideration." (P15, Pregnant, Unvaccinated)

**Individual and group influences**

Individual and group influences reported in this study (Table 2) align with the SAGE model's: 1) personal, family, and community experiences with vaccination; 2) beliefs and attitudes about health and prevention; 3) knowledge and awareness about COVID-19 vaccines; 4) knowledge of the health system and trust in healthcare workers; 5) perceived risks and benefits; 6) and social norms.

**Personal, family, and community experiences with vaccination.** Pregnant and postpartum vaccinated women reported influences from their social environment that seemed to nudge them towards COVID-19 vaccination. This included peers and social media, family members, and healthcare workers, as well as when they saw their neighbors and teachers in their area take up vaccination.

"I discussed it with my sister, and we decided to get vaccinated together for protection." (P17, Pregnant, Vaccinated)

"When we saw other people going for the vaccine, neighbors, teachers in school, we decided to go for it." (A14, Pregnant, Vaccinated)

Pregnant unvaccinated women had consulted healthcare workers, spouses, and other family members, and remained skeptical and hesitant about vaccination. Regarding specific guidance sought from particular contacts, the women inquired about dosage, safety, and side effects from healthcare workers. They discussed vaccine safety with spouses and other family members, while peers and friends discussed safety and provided information on the vaccination recommendations. Unvaccinated pregnant and postpartum women said social networks could discourage COVID-19 vaccination, e.g., through talks on side effects.

"I was influenced by people around me, including what they were saying about side effects, which discouraged me." (A16, Postpartum, Unvaccinated)

"Yes, my friend was telling me 'It is painful you will never wake up for three days,' and that made to fear [vaccination]." (A16, Postpartum, Unvaccinated)

*Reasons for refusing vaccination*

Pregnant and postpartum women reported several reasons for not opting for the COVID-19 vaccine. First, as mentioned above, some had fears about the side effects of the vaccine, including the possibility of infertility, swelling of the arm, and concerns about asthma-related side effects.

"I was told it has a high risk to a person like me who has asthma." (A03, Pregnant, Unvaccinated)

"I heard that it gives erectile dysfunction, and for pregnant women, I heard that it might affect your baby." (A06, Pregnant, Unvaccinated).

"There is a girl we were working with, and when she got injected, her hand swelled, and that's where my fear began." (P13, Postpartum, Unvaccinated)

The unvaccinated also expressed a lack of confidence in the research behind the COVID-19 vaccine. They seemed to hope for more investigations to validate the hurriedly done COVID-19 research. They were concerned about the potential vaccine impact on breastfeeding.

"I don't think I would also agree to it because I feel like it may bring harm to the baby." (A20, Postpartum, Unvaccinated).

"I think there is not much research done for it [COVID-19 vaccine] because things were run in a hurry, and this is where we are." (A15, Postpartum, Unvaccinated)

"I just feel maybe it's not safe... maybe it's my instinct. For me, it's no [to getting vaccinated]." (A12, Postpartum, Unvaccinated)

Secondly, some of the unvaccinated experienced technical hitches as they tried to fulfil the required self-registration online for COVID-19 vaccination.

"The online system was not user-friendly, especially for older people or those who aren't used to such technology." (A09, Pregnant, Unvaccinated)

"Not everyone has internet access or knows how to navigate the online registration. This made getting vaccinated harder for many."(A16, Postpartum, Unvaccinated)

In addition, some of the pregnant and postpartum vaccinated women reported a lack of reliable caregivers who could take care of their children while they went to get vaccinated. It was difficult to make childcare arrangements.

"I had to arrange for someone to look after the children while I went for the vaccine. It wasn't easy to find someone available at that time." (P20, Pregnant, Unvaccinated).

"It was difficult because I had to make sure my kids were safe before I could leave the house to go get vaccinated." (P25, Pregnant, Unvaccinated)

"I struggled to find someone to care for my kids, so I delayed getting the vaccine." (P18, Pregnant, Unvaccinated)

### Reasons for refusing the second dose
Looking at women who had already received a dose of the COVID-19 vaccine, study findings show some hesitancy among pregnant and postpartum women to take the second dose. Postpartum women refused booster doses due to fear of side effects and experiences of negative reactions.

"I am not getting the booster…because of the same concerns - I have a feeling these vaccines were not properly tested." (A15, Postpartum, Vaccinated)

### Experiences with the COVID-19 vaccination and associated processes
Participants reported both positive and negative experiences with vaccination. For vaccinated women reporting positive experiences, the vaccination services were easily accessible within short distances from home at nearby local facilities. For others, the outreach mobile vaccination clinics that went into the local communities were associated with vaccination. They were glad because they did not need to worry about fares to far-off vaccination centers.

"It was easily accessible for me; I just walked to the hospital and requested the vaccine." (A24, Postpartum, Vaccinated)

Once at the vaccination center, they appreciated the smooth flow of the service and received either single-dose or other types of vaccines – it was all dependent on the women's choice and what was available.

"When I went to the center, they had options for vaccines. I received the single-dose one because it was convenient." (P25, Pregnant, Vaccinated)

"It was very easy for me; it was very straightforward. I didn't even have to wait in queues or anything." (A04, Pregnant, Vaccinated)

There were also logistical difficulties and negative experiences reported by women who went for the COVID-19 vaccination. Firstly, similar to the reports by unvaccinated women, some were frustrated as they used the online self-registration process for COVID-19 vaccination.

"It was really hard to register online for the vaccine. The process was slow, and I had to keep re-entering details. It would have been easier if I could just walk in without all that." (A22, Postpartum, Vaccinated)

"Many people, including myself, found it difficult to register online for the vaccine. It was frustrating, and some gave up on getting vaccinated because of the complicated process." (A19, Postpartum, Vaccinated)

For others, the fares needed to get to the health facilities were also an obstacle that they needed to overcome. Moreover, some women found long queues and had to wait for a prolonged time before getting vaccinated.

"The long queues, yeah...The first dose was very hard to get because there were so many people." (A07, Pregnant, Vaccinated)

Lastly, one vaccinated pregnant woman reported experiencing pain after vaccination; however, this did not seem to bother her extensively, as the discomfort lasted for a day.

"The first vaccine was not painful when being injected, but after some time, it would feel numb, and some pain, and the hand would become heavy for one day." (A13, Pregnant, Vaccinated)

**Beliefs and attitudes about health and prevention.** The pregnant and postpartum vaccinated women held positive beliefs and attitudes about health and COVID-19 prevention. As reported under contextual factors, they believed they were all at risk of contracting COVID-19. They knew it was a novel virus and feared contracting the disease. Although they had safety concerns, they believed the vaccine held preventive value and could protect women and their babies. Similarly, the unvaccinated pregnant and postpartum women's beliefs and attitudes about health and prevention revealed an appreciation of the risk of getting infected with COVID-19, and they feared contracting the disease.

"The immunity is lower during pregnancy, and I feared catching COVID because it's easy to get sick." (A06, Pregnant, Unvaccinated)

"I believe the vaccine is good for protection; it will prevent COVID and also protect the baby." (P25, Pregnant, vaccinated)

**Knowledge and awareness about COVID-19 vaccines.** On knowledge and awareness about COVID-19 vaccines, vaccinated and unvaccinated participants reported two factors that seemed to support vaccination. Firstly, as reported

above, there were public announcements, outreaches, and communication from healthcare workers to educate patients and community members on the benefits of COVID-19 vaccines. Secondly, pregnant and postpartum women were aware of COVID-19 vaccination services and the diverse types of vaccines available. For instance, some participants had heard of Pfizer, Moderna, and Johnson and Johnson vaccines.

> "I have heard of Pfizer, Moderna, and Johnson and Johnson vaccines at the clinic when I went for my check-up." (A20, Postpartum, Vaccinated)

**Knowledge of the health system and trust in healthcare workers.** On knowledge of the health system and trust in healthcare workers, as noted above, reports from vaccinated women showed knowledge of available COVID-19 vaccines and awareness of vaccination services. Furthermore, the pregnant and postpartum vaccinated women reported trust in the healthcare worker's recommendations on the choice of vaccine.

> "I trusted the nurse's advice when she recommended which vaccine to take because she explained it well." (A24, Postpartum, Vaccinated)

> "I think I didn't have much say about it. When the doctor recommended it, my only concern was 'Is it safe for pregnancy?' That's all, then I took it." (A20, Postpartum, Vaccinated).

Looking at the unvaccinated participants, the findings on knowledge and awareness about COVID-19 vaccines, knowledge of the healthcare system, and trust in healthcare workers were similar to those of the vaccinated women. The unvaccinated were aware of COVID-19, vaccination services, and health messaging on the benefits of COVID-19 vaccines.

**Perceived risks and benefits.** The SAGE model's determinant of perceived risks versus benefits was captured in reports by both vaccinated and unvaccinated pregnant and postpartum women. As noted above, they feared the vaccine's side effects but appreciated its protective value on women and their babies. Additionally, as indicated above, women with potential complications or underlying health conditions were likely to be cautious about vaccination.

**Social norms.** The final set of factors reflected social norms. Vaccinated and unvaccinated pregnant and postpartum women reported general acceptance of COVID-19 vaccines as essential due to the threat of COVID-19 recurrence and risk of exposure. There was social pressure to get vaccinated because the vaccine was seen as indispensable for the overall well-being of vulnerable persons and other community members. For instance, employers were keen to have all workers vaccinated to avoid/reduce work-related COVID-19 risks.

> "The community feels the vaccine is needed for everyone's safety, especially to protect the vulnerable." (A01, Postpartum, Unvaccinated)

> "Most people here believe in the vaccine because COVID keeps coming back, and you never know who might be infected." (P21, Pregnant, Vaccinated)

> "My employer was like, 'You get the vaccine, or you don't come to work.'" (A15, Postpartum, Unvaccinated)

Nonetheless, pregnant and postpartum vaccinated women had noted reports of laxity with vaccination over time because community members began to perceive COVID-19 vaccination as unnecessary, as they observed a low number of COVID-19 cases or deaths.

> "COVID is no longer a problem right now because people have learnt how to deal with it, and again the cases have gone down, not like before." (A16, Postpartum, Unvaccinated)

## Vaccine-specific factors

The study findings show one vaccine-specific influence from the SAGE model: scientific evidence on benefits versus risks (Table 2). Vaccinated and unvaccinated pregnant and postpartum women discussed limitations in scientific knowledge on COVID-19 vaccines.

> "I think the vaccine is not a hundred percent safe, and there's not much research done. I have my reservations about what it contains." (A20, Postpartum, Vaccinated)

> "... [COVID-19 vaccine] was not like other vaccines, where you get years and years of study and time for production. So, this one, because of the short production time, of course, it's limited in terms of following up with people and seeing how the vaccine has affected their lives." (P13, Postpartum, Unvaccinated)

Vaccinated and unvaccinated participants also discussed the benefits of COVID-19 vaccination; it was valued for its protective effect that consequently enhanced the health and well-being of both mother and child.

## Vaccination-specific factors

The study findings show five vaccination-specific issues (Table 2), including 1) mode of administration, 2) design of the vaccination program, 3) reliability and source of vaccine supplies, 4) vaccination schedule, and 5) the strength of healthcare workers' recommendations.

**Mode of administration.**  According to unvaccinated participants, the mode of administration seemed to scare both pregnant and postpartum women from taking up vaccination because there were fears about the pain associated with the COVID-19 injection. Moreover, there were reports of numbness and swelling of the injected hand that compounded their fears.

> "For the COVID-19 vaccine, I have not received it yet. That is because I fear syringes… I can [confidently] go get the vaccine because it prevents me from COVID-19, but I have the fear of syringes." (A09, Postpartum, Unvaccinated)

**Design of the vaccination program.**  As noted under contextual and individual/group influences above, a program that included community COVID-19 vaccination outreaches seemed to enhance reach as the service was geographically accessible to many members within the locality of the mobile service. Easily accessible vaccination points seemed to encourage people to get vaccinated.

> "The community drive was effective because they were giving the vaccine in community centers, which made it easy for people to get vaccinated." (A04, Pregnant, Vaccinated).

**Source of supplies.**  The availability of vaccines seemed to hearten vaccination. Where the vaccines were readily and routinely supplied, women seemed encouraged to utilize them.

> "Yes, I received the vaccine because it's provided here regularly at the clinic. They have been consistent with offering it." (P22, Pregnant, Vaccinated)

> "It [getting the COVID-19 vaccine] was easy, the only challenge was they ran out at some point, so we had to wait beyond the recommended time to get the injection." (P07, Pregnant, Vaccinated)

Similar to findings among the vaccinated participants, unvaccinated pregnant and postpartum women generally reported the availability of vaccines and related supplies.

"The government has tried by making vaccines available even in small clinics. They have provided masks and sanitizers as well." (P24, P, Unvaccinated)

However, some of the unvaccinated women had limited knowledge of vaccine availability. For instance, some of the pregnant unvaccinated women believed the vaccine was not currently available, while others did not know which specific COVID-19 vaccines were available. Furthermore, some of the postpartum women were unsure of the availability of the COVID-19 vaccine for recently pregnant or breastfeeding.

**Vaccination schedule.** According to vaccinated participants, a simple, straightforward schedule seemed to encourage people to utilize the vaccination service. For instance, findings showed a preference among both pregnant and postpartum women for single-dose vaccines, which was unsurprising because some participants had complained about the painful injection in their arm.

"For COVID, I got the Johnson and Johnson vaccine... I chose it because it was a single dose." (A02, Postpartum, Vaccinated)

**Healthcare workers' recommendations.** According to vaccinated and unvaccinated participants, the trust pregnant and postpartum women had in healthcare workers' recommendations regarding COVID-19 vaccines could affect vaccination uptake. Furthermore, findings from vaccinated and unvaccinated women suggest there was limited information on COVID-19 vaccinations for pregnant women. There seemed to be a need for intentional healthcare workers' engagement with their clients on the importance of getting vaccinated.

"Nothing much [heard] about [COVID-19 vaccine recommendations for] the pregnant women" (A11, Pregnant, Vaccinated)

"I have not heard anything [on COVID-19 vaccine recommendations] from doctors or community health workers" (A17, Postpartum, Vaccinated)

"No, I have not heard anything [on specific guidance on COVID-19 vaccination for pregnant women from healthcare workers]" (P09, Postpartum, unvaccinated).

Study findings suggest that trust in healthcare workers' recommendations regarding COVID-19 vaccines can nudge women to vaccination.

"The doctor told me it was safe and recommended I take the Pfizer vaccine; I believed him and took it." (P25, Pregnant, Vaccinated)

## Discussion

This study provides qualitative findings on pregnant and postpartum women's attitudes, beliefs, and behaviors related to COVID-19 vaccination during pregnancy in Kenya. The COVID-19 pandemic has highlighted the challenges of maternal vaccine acceptance and the need to investigate factors associated with vaccination in this population [2–5].

### Contextual influences *on* COVID-19 vaccination

Our study shows that information sources, content, and risk communication are all-important for pregnant women in Kenya and may influence the uptake of maternal vaccines, such as COVID-19, and the use of follow-up boosters. Communities with access to the mainstream media are more likely to access information and accept the COVID-19 vaccine

than those who lack it [2]. Vaccine refusal is fueled by misinformation and disinformation on social media platforms about the severity, efficacy, and safety of the vaccine [2,18,37–39]. Furthermore, as noted in this study, mistrust in government motives can dissuade COVID-19 vaccination uptake, with hesitancy noted among pregnant women who do not trust the government as a source of COVID-19 information [40]. For instance, in Kenya, the government was accused of under-taking the COVID-19 vaccination to get external funding, which heightened mistrust in vaccinations [18,21]. A lack of trust in government agents and their health development partners implies a reliance on untrustworthy sources that spread misinformation. Practical, tailored initiatives can enhance trust in government vaccination programs. Given the association of lack of trust in health authorities with prior vaccinations, colonial vestiges, misinformation, and inadequate knowledge about vaccine efficacy [41–43], nationwide campaigns and effective communication on COVID-19 vaccines can enhance trust in government vaccination efforts. Noteworthy, efforts that bring out the clear vaccination benefits for both pregnant women and their infants are likely to encourage complete/proper immunization [7].

In Kenya, a lack of eligibility information has been cited as a barrier to uptake by pregnant women [23]. The absence of clear national policy guidance on COVID-19 vaccines influences behaviors in pregnant and breastfeeding women, and this calls for the development of relevant policies and campaigns that prioritize this subpopulation, including effective communication guidelines to disseminate vaccination information across the nation [13,23,26]. Globally, failure of national vaccine recommendations to consistently include guidelines on COVID-19 vaccines for pregnant women might also have an impact on vaccine uptake. For instance, in 2021, only 10 out of the 54 countries in Africa recommended COVID-19 vaccines for some or all pregnant women [44].

Furthermore, recommendations by the government and other international policymakers, such as the WHO, can enable uptake of the COVID-19 vaccine. In line with the WHO recommendations, policymakers and programs should prioritize pregnant women for COVID-19 vaccination [14,16,44].

### Individual and group influences on COVID-19 vaccination

On individual and group influences, healthcare workers were consulted by both the vaccinated and unvaccinated women. Past research shows that health workers are considered the most trusted sources of guidance about COVID-19 vaccine choices [2,18,21,36,39,43,45]. Pregnant, unvaccinated women also consulted spouses and other family members. Moreover, both pregnant and postpartum vaccinated women reported reliance on friends and social media. Our study reinforces earlier findings from Kenya that indicate proximal social networks, e.g., family members, can play a key role in dissuading pregnant women from COVID-19 vaccination [46].

Vaccine uptake is dependent on individual and social factors, as it is influenced by age, sex, marital status, education level, employment status, religion, geographical location, and knowledge of someone who has tested positive for COVID-19 [2,18,21,36,39,43,45]. Similar to this study, vaccine hesitancy has been attributed to fear of adverse events, such as pain at the injection site [15], and side effects [16,18]. As noted above, any fake news on side effects and miscommunication can discourage vaccination. The women in our study were aware of the side effects, but they also had information about the benefits of COVID-19 vaccination, and they knew where to get vaccination services. This implies that the health campaigns, government announcements, and other messaging forums had borne fruit.

Our study participants reported both pragmatic and non-pragmatic reasons for their positive and negative experiences as they sought COVID-19 vaccination. Noteworthy, Kenya recommended COVID-19 vaccination for all pregnant and breast-feeding women from 2023 [14], and our data collection occurred between October 2023 to March 2024. This implies that the policy shift may have reduced barriers to pregnant women who wanted to get a COVID-19 vaccination. Vaccinated women were happy with the easily accessible vaccination services offered smoothly. On negative experiences, both pregnant and postpartum women reported difficulties associated with childcare while they went for vaccination, long service queues, and pain at the injection site. Some unvaccinated women complained about the fares to far-off vaccination services. Appreciating these experiences provides an opportunity to design and deliver maternal vaccines in a more user-friendly manner.

Our study participants reported that COVID-19 remains a persistent public health threat, which can be partly overcome through the use of the COVID-19 vaccine. Past studies show that underestimation of disease severity, misinformation about COVID-19, and suggestions that a cure is attainable through safer alternative treatments can discourage uptake of the vaccine [47,48]. Similar to our findings, past studies suggest that social pressure within communities makes decision-making in the uptake of the COVID-19 vaccine a response to subjective norms [20]. The desire to be protected from COVID-19 and a concern for the well-being of others to whom individuals might transmit COVID-19 encourage vaccination; therefore, messaging that focuses on the benefits to the health of other people, such as family, the vulnerable, and friends, is a positive driver to vaccine uptake [49].

## Vaccine-specific influences

The vaccine-specific influences noted in our study were reported by vaccinated and unvaccinated participants. They had heard of women experiencing numbness and swelling of the injected hand, and both pregnant and postpartum women expressed fears about the painful injection method. Other influences included the availability of COVID-19 information on the vaccine, safety assurance, and effectiveness in protecting mother and child. Low uptake of COVID-19 vaccines has been associated with doubts about the safety and efficacy of the vaccines [18,21,23,34,47]. Unsurprisingly, fast development and rollout of new vaccines after a pandemic breakout can attract vaccine hesitancy because people fear the unknown [18,20,34]. Low uptake of some maternal vaccines can also be due to inadequacies in safety data, inaccuracies regarding potential fetal harm [18,23]. Furthermore, in the context of Africa, there have been past reports on vaccine scandals, including reports on critical side effects of vaccination [50].

Initiatives that bypass continental efforts inadvertently compound the mistrust by breeding allegations that Africa is a testing ground for vaccines [50]. For example, a Cameroonian study revealed a deep mistrust of the pharmaceutical companies producing the vaccine, as there was a belief that they were aiming for financial gain instead of public health interests [51]. While diverse factors continue to discourage vaccine uptake, all stakeholders must continue working towards proper provision of the vaccine to pregnant women, as past evidence shows its clear benefits for pregnant women and their infants [7].

## Vaccination-specific influences

This study reported COVID-19 vaccination-specific facilitators, including vaccination outreaches and availability of different types of vaccines. Service design (e.g., the distance to vaccination service points) has implications on vaccination service experience for clients, and transportation needs that pregnant women have to consider. Difficulties accessing vaccines, captured in costs (e.g., fares) incurred, inconveniently timed and far-off service locations, crowded services, and vaccine stock-outs can all discourage the uptake of vaccination services [23,32,36,48].

In this study, both pregnant and postpartum women expressed a preference for single-dose vaccines, which are conveniently dealt with in one trip. Findings also show that vaccination-specific facilitators are associated with access to trusted healthcare workers who freely provide advice on vaccination. The success of any vaccination program is largely determined by the level of trust in the vaccines as well as the institutions that administer them [52]. Trust and willingness to be vaccinated are strongly connected, and where doctor-patient trust exists, the providers are powerful agents in allaying fears and consequently shaping patients' vaccination behaviours [53]. Given the confirmed value of complete or boosted vaccination status among pregnant women (e.g.,  reduced risks of morbidity and complications), government officials, healthcare workers, and other stakeholders ought to intentionally invest in expanded vaccination coverage [6].

## Strengths and limitations

The strengths of this study are the use of two sites and diverse categories of participants, including women at different gestational ages. This allowed for the generation of useful descriptions needed to understand factors associated with

COVID-19 vaccination among pregnant women. To ensure rigor, two research team members coded the data, and all investigators discussed the themes to reach consensus. On study limitations, we only included pregnant and postpartum women, who already had some knowledge about the COVID-19 vaccine, and therefore did not include the voices of other stakeholders who may influence maternal vaccination (e.g., spouses and healthcare workers). Given the topic of vaccination, social desirability may have made the women answer questions in a manner that could be viewed favorably by the research team. Furthermore, this study was undertaken in the post-pandemic period, which might influence recall and associated reporting on past experiences.

While results from this study may not be applicable outside of the study setting, the factors reported as potentially influencing COVID-19 vaccination among pregnant women may apply to other similar contexts in Kenya and beyond. Despite the study's limitations, it expands our understanding of factors associated with COVID-19 vaccination among pregnant women, especially in the Kenyan context. It specifically provides information on vaccine hesitancy determinants that can be investigated further in future studies.

## Conclusion

Our qualitative study revealed an array of factors (contextual, individual-level, group-level, vaccine-specific, and vaccination-specific issues) that potentially influence COVID-19 vaccination among pregnant women in Kenya. These findings may apply to similar contexts globally and thus provide useful insights into vaccine hesitancy. Findings can be used to inform other maternal vaccine initiatives, such as the Respiratory Syncytial Virus (RSV) vaccine and the upcoming Group B Streptococcus (GBS) vaccine. Future research should apply more rigorous research approaches to compare factors influencing the uptake of COVID-19 vaccines by pregnant versus postpartum women and vaccinated versus unvaccinated women. Furthermore, future studies should examine the long-term consequences of these pregnancy decisions to inform more contextually relevant counseling in this population. This is important because women ought to be fully informed and persuaded towards vaccination, given the extensive benefits it brings to pregnant women and their infants. In addition, our study suggests vaccination programming that addresses physical, psychological, social, and economic factors impacting this population can support their unique needs. For maximum impact, such programming should be tailored and segmented to address pregnant women, postpartum women, and those living in low-resource settings.

## Supporting information

**S1 File. SAGE vaccine hesitancy matrix of determinants.**
(DOCX)

**S2 File. Semi-structured interview guides.**
(DOCX)

**S1 Checklist. COREQ (COnsolidated criteria for REporting Qualitative research) Checklist.**
(PDF)

**S2 Checklist. Inclusivity in Global Research.**
(DOCX)

## Acknowledgments

The authors wish to acknowledge the support received from Rupali J. Limaye, Jessica L. Schue, Grace Belayneh, Vanessa Brizuela, regional health officials, health facility administrators, research ethics committees, research assistants, and study participants. This work was conducted in collaboration with the Johns Hopkins Bloomberg School of Public Health, UNDP-UNFPA-UNICEF-WHO-World Bank Special Programme of Research, Development and Research Training

in Human Reproduction (HRP), a cosponsored programme executed by the World Health Organization (WHO) [HRP Project ID: A66042], The Aga Khan University (Pakistan), The Aga Khan University (Kenya), University of Ghana, and State University of Campinas (Brazil).

## Author contributions

**Conceptualization:** Ferdinand Okwaro, Marleen Temmerman.

**Data curation:** Violet Naanyu, Ferdinand Okwaro.

**Formal analysis:** Violet Naanyu, Ferdinand Okwaro.

**Investigation:** Ferdinand Okwaro, Ingrid Gichere, Mandeep Sura, Prachi Singh, Berhaun Fesshaye, Marleen Temmerman.

**Methodology:** Violet Naanyu, Ferdinand Okwaro, Ingrid Gichere, Mandeep Sura, Prachi Singh, Berhaun Fesshaye, Marleen Temmerman.

**Project administration:** Ferdinand Okwaro, Marleen Temmerman.

**Supervision:** Marleen Temmerman.

**Validation:** Violet Naanyu, Ferdinand Okwaro, Ingrid Gichere, Mandeep Sura, Prachi Singh, Berhaun Fesshaye, Marleen Temmerman.

**Visualization:** Violet Naanyu.

**Writing – original draft:** Violet Naanyu, Ferdinand Okwaro, Ingrid Gichere, Mandeep Sura.

**Writing – review & editing:** Violet Naanyu, Ferdinand Okwaro, Ingrid Gichere, Mandeep Sura, Prachi Singh, Berhaun Fesshaye, Marleen Temmerman.

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
