## [Decision Letter · Decision Letter 0]

5 Oct 2025

PGPH-D-25-02543

Qualitative exploration of factors associated with COVID-19 vaccination among pregnant women in Kenya

Dear Dr. Naanyu,

Thank you for submitting your manuscript to PLOS Global Public Health. After careful consideration, we feel that it has merit but does not fully meet PLOS Global Public Health’s publication criteria as it currently stands. Therefore, we invite you to submit a revised version of the manuscript that addresses the points raised during the review process.

We look forward to receiving your revised manuscript.

Kind regards,

Paolo Ivo Cavoretto

Academic Editor

Journal Requirements:

1. Please provide a detailed online Financial Disclosure statement. This is published with the article. It must therefore be completed in full sentences and contain the exact wording you wish to be published.

a) Please clarify all sources of financial support for your study. List the grants, grant numbers, and organizations that funded your study, including funding received from your institution. Please note that suppliers of material support, including research materials, should be recognized in the Acknowledgements section rather than in the Financial Disclosure.

b) State the initials, alongside each funding source, of each author to receive each grant. For example: “This work was supported by the National Institutes of Health (####### to AM; ###### to CJ) and the National Science Foundation (###### to AM).”

c) State what role the funders took in the study. If the funders had no role in your study, please state: “The funders had no role in study design, data collection and analysis, decision to publish, or preparation of the manuscript.”

For more information, please go to our submission guidelines:

https://journals.plos.org/globalpublichealth/s/submission-guidelines#loc-financial-disclosure-statement

2. Please ensure that the funders and grant numbers match between the Financial Disclosure field and the Funding Information tab in your submission form. Note that the funders must be provided in the same order in both places as well.

3. In this instance it seems there may be acceptable restrictions in place that prevent the public sharing of your minimal data. However, in line with our goal of ensuring long-term data availability to all interested researchers, PLOS’ Data Policy states that authors cannot be the sole named individuals responsible for ensuring data access (http://journals.plos.org/globalpublichealth/s/data-availability#loc-acceptable-data-sharing-methods).

Additional Editor Comments (if provided):

Please revise accordingly. Please consider the INTERCIVID experience it is a multinational consortium producing major evidence on this end.

Reviewers' comments:

Reviewer's Responses to Questions

**Comments to the Author**

1. Does this manuscript meet PLOS Global Public Health’s publication criteria?

Reviewer #1: Yes

Reviewer #2: Yes

2. Has the statistical analysis been performed appropriately and rigorously?

Reviewer #1: Yes

Reviewer #2: N/A

3. Have the authors made all data underlying the findings in their manuscript fully available (please refer to the Data Availability Statement at the start of the manuscript PDF file)?

Reviewer #1: Yes

Reviewer #2: Yes

4. Is the manuscript presented in an intelligible fashion and written in standard English?

Reviewer #1: Yes

Reviewer #2: Yes

Reviewer #1: General comment: This is a valuable, well-motivated study with strong fieldwork and clear relevance for maternal immunization programs.

More details are provided in the attachment

Reviewer #2: This research article entitled “Qualitative exploration of factors associated with COVID-19 vaccination among pregnant women in Kenya” provides valuable information pregnant and postpartum women in relation with COVID-19 vaccination. The study design and methodology are robust and address an important public health issue. The manuscript is well-written. However, here are some comments to improve their quality:

1. The manuscript needs a further literature review to include current evidence on this topic.

For example, there are other COVID-19 vaccine studies conducted with pregnant and postpartum women in Kenya, that have not been referenced, such as:

Marwa MM, et al. COVID-19 vaccine hesitancy among pregnant and postpartum Kenyan women. Int J Gynaecol Obstet. 2023 Jul;162(1):147-153. doi: 10.1002/ijgo.14773. Epub 2023 Apr 10. PMID: 37036449; PMCID: PMC10330087.

Ayieko S, Jaoko W, Opiyo RO, Orang'o EO, Messiah SE, Baker K, Markham C. Knowledge, Attitudes, and Subjective Norms Associated with COVID-19 Vaccination among Pregnant Women in Kenya: An Online Cross-Sectional Pilot Study Using WhatsApp. Int J Environ Res Public Health. 2024 Jan 16;21(1):98. doi: 10.3390/ijerph21010098. PMID: 38248561; PMCID: PMC10815556.

2. Methods: Why didn’t you include interviews with the partners? I understand that is key to have women’s perspectives, but children care is responsibility of both parents (in case of pregnant women who are within a relationship, that one could guess that is most of cases). Partners’ views, as well as family views, are key for decision-making in vaccine acceptability for pregnant women. This is well reflected in the limitations section.

3.Results: Please, review and strengthen the first paragraph of the Results section.

4. Results: I would encourage authors to include a “Table 1” with sociodemographic characteristics of participants in the study. It would be valuable to see not only the basic sociodemographic factors, but the number of women who accepted COVID-19 and other vaccines during pregnancy.

5. Results: Within a quote you included the “HCP” acronym, however it was not been defined, nor used throughout the manuscript. I would encourage you to use the acronym HCW or HCP, if not, to please, delete that and include the full definition of the term.

6. Discussion: It would be worth to discuss a bit more the role of the fake news, and miscommunication. For example, some participants declared some fears that lead to non vaccination, such as vaccination affecting erectile function, fertility, population control etc.

7. Results: I would have expected some results discussing the role of spirituality, religion, or religious leaders in vaccine acceptance or hesitancy, as it was briefly mentioned in the discussion section that it was one of the factors affecting other studies. Did you explore this within the data?

Hope these comments help improve the manuscript.

Sincerely,

**Do you want your identity to be public for this peer review?** For information about this choice, including consent withdrawal, please see our Privacy Policy

Reviewer #1: No

Reviewer #2: **Yes:** Dr Elena Marbán-Castro

---

## [Decision Letter · Decision Letter 1]

1 Dec 2025

PGPH-D-25-02543R1

Qualitative exploration of factors associated with COVID-19 vaccination among pregnant women in Kenya

Dear Dr. Naanyu,

Thank you for submitting your manuscript to PLOS Global Public Health. After careful consideration, we feel that it has merit but does not fully meet PLOS Global Public Health’s publication criteria as it currently stands. Therefore, we invite you to submit a revised version of the manuscript that addresses the points raised during the review process.

We look forward to receiving your revised manuscript.

Kind regards,

Paolo Ivo Cavoretto, MD PhD

Academic Editor

Journal Requirements:

Additional Editor Comments:

The revision was successful. However before publication I believe some work should be done to reinforce the concept that vaccine is beneficial for mothers, fetuses and neonates. Particularly the authors may enrich the background with knowledge from major studies showing that neonates from vaccinated mothers have lower risks. Neonates of vaccinated mothers had a decreased risk for preterm birth and adverse neonatal outcomes. (1)

More so the authors should support the effort in preventing vaccine hesitancy as stated in recent opinions and major work (2) Please refer them to major guidelines of other coutries and international consortia if the trust in local authorities is low. Please add this concepts in order to reinforce vaccine uptake locally, this is critical to improve the outcome.

1. Barros FC, et al. ; INTERCOVID-2022 International Consortium. Maternal vaccination against COVID-19 and neonatal outcomes during Omicron: INTERCOVID-2022 study. Am J Obstet Gynecol. 2024 Oct;231(4):460.e1-460.e17. doi: 10.1016/j.ajog.2024.02.008. Epub 2024 Feb 16. PMID: 38367758.

2. Cavoretto PI, Farina A. Time to enhance COVID-19 vaccination in women of reproductive age. Lancet Reg Health Eur. 2024 Sep 10;45:101069. doi: 10.1016/j.lanepe.2024.101069. PMID: 39308776; PMCID: PMC11415629.

Reviewers' comments:

Reviewer's Responses to Questions

**Comments to the Author**

Reviewer #1: All comments have been addressed

Reviewer #2: All comments have been addressed

publication criteria?

Reviewer #1: Yes

Reviewer #2: Yes

3. Has the statistical analysis been performed appropriately and rigorously?

Reviewer #1: Yes

Reviewer #2: N/A

4. Have the authors made all data underlying the findings in their manuscript fully available (please refer to the Data Availability Statement at the start of the manuscript PDF file)?

Reviewer #1: Yes

Reviewer #2: (No Response)

5. Is the manuscript presented in an intelligible fashion and written in standard English?

Reviewer #1: Yes

Reviewer #2: Yes

Reviewer #1: The authors have addressed the comments appropriately.

Reviewer #2: Thank you for addressing the comments from the first round of peer review. My only remaining comment is that you need to update the COREQ checklist. Please make sure to indicate the page numbers in the manuscript where each piece of information is provided, rather than adding explanations or repeating content. For example, in the cell where you listed “NVIVO,” you should instead specify the page number in the manuscript where the use of this software is described. The same applies to items such as the derivation of themes and other methodological details, etc. Please update the checklist accordingly.

**Do you want your identity to be public for this peer review?** For information about this choice, including consent withdrawal, please see our Privacy Policy

Reviewer #1: No

Reviewer #2: **Yes:** Elena Marbán Castro

---

## [Decision Letter · Decision Letter 2]

18 Jan 2026

Qualitative exploration of factors associated with COVID-19 vaccination among pregnant women in Kenya

PGPH-D-25-02543R2

Dear Dr Naanyu,

We are pleased to inform you that your manuscript 'Qualitative exploration of factors associated with COVID-19 vaccination among pregnant women in Kenya' has been provisionally accepted for publication in PLOS Global Public Health.

Best regards,

Julia Robinson

Executive Editor

Reviewer Comments (if any, and for reference):

Reviewer's Responses to Questions

**Comments to the Author**

Reviewer #1: All comments have been addressed

publication criteria?

Reviewer #1: Yes

3. Has the statistical analysis been performed appropriately and rigorously?

Reviewer #1: Yes

4. Have the authors made all data underlying the findings in their manuscript fully available (please refer to the Data Availability Statement at the start of the manuscript PDF file)?

Reviewer #1: Yes

5. Is the manuscript presented in an intelligible fashion and written in standard English?

Reviewer #1: Yes

Reviewer #1: Thank you for your revision.

**Do you want your identity to be public for this peer review?** For information about this choice, including consent withdrawal, please see our Privacy Policy

Reviewer #1: No
